# Repurposing Ferumoxytol as a Breast Cancer-Associated Macrophage Tracer with Five-Dimensional Quantitative [Fe]MRI of SPION Dynamics

**DOI:** 10.3390/cancers13153802

**Published:** 2021-07-28

**Authors:** Laurel O. Sillerud, Alexander J. Neuwelt, Fernanda I. Staquicini, Wadih Arap, Renata Pasqualini

**Affiliations:** 1Department of Neurology, UNM BRaIN Center, University of New Mexico School of Medicine, Albuquerque, NM 87106, USA; 2Division of Hematology, Oncology and Palliative Care, Department of Internal Medicine, Virginia Commonwealth University School of Medicine, Richmond, VA 23298, USA; Alexander.Neuwelt@va.gov; 3Department of Medical Oncology, Veterans Affairs Medical Center, Richmond, VA 23249, USA; 4Rutgers Cancer Institute of New Jersey, Newark, NJ 07103, USA; fstaquicini@mbracetrx.com (F.I.S.); wa116@newark.rutgers.edu (W.A.); 5Division of Hematology/Oncology, Department of Medicine, Rutgers New Jersey Medical School, Newark, NJ 07103, USA; 6Division of Cancer Biology, Department of Radiation Oncology, Rutgers New Jersey Medical School, Newark, NJ 07103, USA

**Keywords:** ferumoxytol, triple-negative breast cancer, MRI, [Fe]MRI

## Abstract

**Simple Summary:**

With the incorporation of immune-modulating therapies into the standard management of triple-negative breast cancer, there is increased interest in the non-invasive imaging of the tumor immune microenvironment. Ferumoxytol is FDA-approved as an iron replacement therapy for iron-deficiency anemia and is also a superparamagnetic iron oxide nanoparticle (SPION) resulting in negative enhancement on T_2_-weighted MR imaging. It has previously been established that ferumoxytol is taken up by macrophages. In the current study, we used ferumoxytol-contrasted MRI to quantitatively image the iron concentration, and, by extension, the tumor-associated macrophage infiltration within the tumor microenvironment of a highly inflammatory model of triple-negative breast cancer.

**Abstract:**

Tumor-associated macrophages (TAMs) in breast cancer regulate inflammation, immunosuppression, angiogenesis, and metastasis. However, TAM imaging remains a clinical challenge. Ferumoxytol has long been an FDA-approved superparamagnetic iron oxide nanoparticle (SPION) preparation used as an intravenous (IV) treatment for iron-deficiency anemia. Given its high transverse relaxivity, ferumoxytol produces a negative image contrast upon cellular uptake in T_2_-weighted magnetic resonance imaging (MRI) studies. Here we evaluated ferumoxytol as a contrast agent to image/quantify TAMs in an aggressive mouse model of breast cancer: We developed [Fe]MRI to measure the 5-dimensional function c(x,y,z,t), where c is the concentration of nanoparticle iron and {x,y,z,t} is the 4-dimensional set of tumor space-time coordinates. Ferumoxytol SPIONs are readily phagocytosed (~10^4^/cell) by the F4/80^+^CD11b^+^ TAMs within breast tumors. Quantitative [Fe]MRIs served to determine both the spatial and the temporal distribution of the SPION iron, and hence to measure [Fe] = c(x,y,z,t), a surrogate for TAM density. In single-dose pharmacokinetic studies, after an IV dose of 5 mg/Kg iron, [Fe]MRI measurements showed that c(x,y,z,t) within breast tumors peaked around [Fe] = 70 μM at 42 h post-administration, and decayed below the [Fe]MRI detection limit (~2 μM) by day 7. There was no SPION uptake in control organs (muscle and adipose tissue). Optical microscopy of tissue sections confirmed that F4/80^+^CD11b^+^ TAMs infiltrated the tumors and accumulated SPION iron. Our methodology and findings have translational applications for breast cancer patients.

## 1. Introduction

Advances in imaging technologies have enhanced the diagnosis, monitoring, and response assessment of breast cancer patients. Yet, breast cancer remains the second most common malignant tumor type worldwide and a leading cause of death among women, with an incidence of ~250,000 new cases per year in the USA [1]. Of those, triple-negative breast cancer (TNBC) comprises up to ~10–20% of all cases. Despite optimal management, many TNBC patients have distant metastases and poor disease outcomes. A biological hallmark of TNBC is an immunosuppressive tumor microenvironment that fosters tumor growth and metastatic spread through the suppression of tumor-infiltrating lymphocytes (TIL) and secretion of inhibitory cytokines, mainly by tumor-associated macrophages (TAM). TAMs may classically be divided into two major populations, termed M1 and M2, representing the extremes of a broad activation state spectrum; the M1 population is associated with anti-tumor activity while the M2 population with tumor progression [2,3,4].

In solid tumors, tissue resident and bone marrow-derived infiltrating macrophages represent a major immune component of the tumor microenvironment [5] and enhance invasive growth, matrix remodeling, angiogenesis, and immune suppression [6,7]. Consistent with their pro-tumorigenic phenotypes, the accumulation of tumor-associated macrophages (TAMs) in breast cancer patients is generally associated with poor prognosis, resistance to therapies, and disease recurrence [8,9,10]. Given the significant clinical impact of macrophage density in solid tumors, developing novel non-invasive imaging methods to quantify TAMs has a clear applicative use in breast cancer patients, particularly given the recent incorporation of immune-modulating therapies, such as checkpoint inhibitors, into standard management of these patients [11]. Driven partly by recent successful attempts to use non-invasive magnetic resonance imaging (MRI) methods to detect inflammation in osteomyelitis [12,13,14,15] and to observe TAMs in murine tumor models [16,17], we sought to develop methods for the magnetic labeling and quantitative detection of TAM-associated Fe in a mouse breast tumor model.

Ferumoxytol is an intravenous (IV) superparamagnetic iron oxide nanoparticle (SPION) preparation currently approved by the U.S. Food and Drug Administration (FDA) for the treatment of iron-deficiency anemia patients, including—but not limited to—those suffering from chronic kidney disease. Ferumoxytol (marketed as Feraheme^®^) has also been studied as an MRI contrast agent in off-label applications [15,18]. Earlier generation SPION formulations have been shown to accurately detect metastatic prostate cancer in lymph nodes, and to outperform conventional MRI by 35–91% [19,20]. Ferumoxytol is phagocytosed by macrophages within tumors to produce persistent signal attenuation in post-contrast T_2_-weighted, and T_2_*-weighted MRI; previous applications of ferumoxytol-contrasted MRI have shown that macrophages can be imaged [12,13,14,15,21,22,23] although the dynamics of the SPION uptake have not yet been examined. In an established breast cancer model, MMTV-PyMT, based on the mouse mammary tumor virus, intraperitoneal (IP) administration of ferumoxytol was taken up selectively by TAMs, and not tumor cells, leading to a signal decline in T_2_-weighted MRI sequences at 24 h post-infusion [24].

Here, we report the translational application of our unique, quantitative magnetic resonance imaging protocol for iron concentration [25,26], hereafter termed [Fe]MRI, in order to image iron concentration, [Fe], and to measure the spatial distribution and temporal duration of macrophage-associated ferumoxytol iron in an experimental syngeneic, mouse model of aggressive triple-negative breast cancer (TNBC), with a strong and well-defined TAM component.

## 2. Materials and Methods

### 2.1. Cells, Animals and Tumor Model

EF43.*fgf-4* tumor cells [27] were grown in standard tissue culture at 37 °C in 5% CO_2_ in Dulbecco’s minimal essential medium supplemented with 10% fetal bovine serum (FBS), penicillin (100 units/mL), and streptomycin (100 μg/mL). For tumor implantation, cells were collected, counted, centrifuged, and re-suspended in serum-free medium. The EF43.*fgf-4* cells (5 × 10^6^) were injected subcutaneously (SC) into either the right flank or the mammary fat pad of syngeneic 4-week-old female BALB/c mice (Harlan-Sprague-Dawley). A uniform cohort of tumor-bearing mice (*n* = 22) was used. Tumor growth was monitored twice a week by caliper measurement and is expressed as the mean tumor volume ± standard deviation (SD). The time-dependent tumor volume data were fitted using Mathematica^®^ (Wolfram Inc., Champagne Urbana, IL, USA) to an exponential growth model, V(t) = 0.092e^1.262t^, which gave a cell doubling time of 19.2 h. Tumors were allowed to develop to a final volume of ~500 mm^3^. The stock ferumoxytol solution (AMAG Pharmaceuticals; Cambridge, MA, USA) containing Fe at a concentration of [Fe] = 30 mg/mL, was diluted 10-fold with phosphate-buffered saline (PBS), and 30–45 µL (90–125 µg Fe) were injected into the tail vein of each mouse according to their individual weights at a dose of 5 mg/kg. All animal procedures were approved by the local Institutional Animal Care and Use Committee and were performed in accordance with all current regulations of the U.S. Department of Health and Human Services.

### 2.2. Determination of the MR Relaxivity of Ferumoxytol at 4.7 Tesla

MRI was performed by using a 72 mm diameter quadrature body coil in a 4.7 Tesla Bruker Biospin (Billerica, MA, USA) MRI system with Avance III electronics running Paravision 5.1 software. The use of [Fe]MRI to measure iron concentration (which we hereby denote with the standard chemistry element concentration notation as [Fe]) required knowledge of the relaxivity of the SPIONs [25,26]. This parameter was determined through the construction and MR imaging of phantoms made from 1% low-melting agarose (Calbiochem, Temecula, CA, USA) in PBS, containing various concentrations of ferumoxytol to produce ten [Fe] values ranging from 0 to 1.0 mM. The longitudinal and transverse relaxation rates of the water in these samples were determined by using Bruker’s standard inversion-recovery and multiple spin-echo sequences and curve-fitting in Paravision 5.1. After the MR measurements were completed, the samples were digested at 90 °C overnight (ON) in 6 N HCl. The [Fe] was determined by using the ferrozine assay by spectrophotometry at 562 nm (Shimadzu, Biospec Mini, Kyoto, Japonia). Standard curves (*n* = 10) were prepared by using FeCl_3_ solutions (Sigma-Aldrich; St. Louis, MO, USA) and provided an average slope of 26,760 ± 928 L/(mole cm) (mean ± SD; COV = 3.5%). The corresponding longitudinal, r_1_, and transverse, r_2_, relaxivities of ferumoxytol were determined from linear fits to plots of the relaxation rate enhancements for water as a function of [Fe].

### 2.3. Animal MRI Measurements

During the MRI measurements, the tumor-bearing mice were kept under deep anesthesia (2% isoflurane in oxygen). Their respiratory rates and rectal temperatures were monitored continuously with the aid of a small animal monitoring system (SAI Inc.; Stony Brook, NY, USA), which also maintained their temperature at 37 °C ± 0.2 °C. Baseline measurements (pre-injection) were acquired at day 0, and tumor volumes and the dynamics of the contrast enhancement in tumors were acquired at 1, 2, 3, 4, and 7 days post-injection. T_1_-weighted (T_1_w) spin-echo images were acquired with a slice thickness of 1 mm, using 256 × 256 pixels of 0.156 mm each, 12 slices, T_R_ = 500 ms, T_E_ = 12.5 ms, and nex = 2, while T_2_-weighted (T_2_w) images used the same parameters except T_R_ = 2500 ms, T_E_ = 50 ms. RARE images were taken with T_R_ = 5000 ms, T_E_ = 14 ms. Since [Fe]MRI also requires knowledge of the pre-contrast relaxation times (T_1_ and T_2_) of the tissue(s) of interest, these were measured by using standard Bruker pulse sequences (RAREVTR, MSME) with variable echo and recycle times. The psoas muscle was chosen as the internal signal intensity standard because it was easily visible in all the axial MR images. We confirmed its suitability as an intensity standard by measuring the slice-to-slice, and animal-to-animal variation from 22 sets of images of eight slices each and found a coefficient of variation of 8.8%.

### 2.4. Contrast Measurement

The primary data obtained consisted of T_1_w and T_2_w MR image intensities, which were used to compute the 5-dimensional quantitative tumor/muscle contrast, C_i_(x,y,z,t), i ∈ {1,2} defined as:C_i_ = (S_iT_ − S_iM_)/S_iM_(1)
where S_iT_(x,y,z,t) was the MR signal of the tumor, and S_iM_(x,y,z,t) was the MR signal of the psoas muscle (Figure 1). These signals were measured both before and after ferumoxytol injection with both T_1_-weighted (T_1_w; i = 1) and T_2_-weighted (T_2_w; i = 2) imaging protocols. We measured image contrast because this was determined by the relative relaxation rates for the tissue water and was independent of the overall image intensity scaling. In general, contrast, as defined here, varies from −1 to +∞, where C = −1 means that the tumor was completely black; i.e., the MR signal was zero from the tumor. As the tumor signal increased with respect to the muscle signal, the contrast increased from negative values to zero as the tumor and muscle reached isointensity. As the tumor signal became larger than that from the muscle, the contrast became positive. The contrast was measured as a function of both the repetition time (T_R_) and the echo time (T_E_) in order to find the optimum MR imaging parameters and to compare these optima with predictions based on the measured relaxation times. The echo time for optimum contrast for T_2_w images was predicted to be [28]:(2)$$t_{\mathrm{opt}}=-\frac{T_{2T}T_{2M}\mathrm{Ln}\left(\frac{T_{2M}}{T_{2T}}\right)}{\left(T_{2T}-T_{2M}\right)}$$
where T_2T_ (T_2M_) was the transverse relaxation time for the tumor (muscle).

### 2.5. Conversion of MRI Contrast Measurements to [Fe]

We converted these contrast measurements to iron concentration ([Fe], mmol/liter) within the tumors by using our novel [Fe]MRI technique [25,26]. In brief, the signal intensity, S_1_(c) in T_1_w, and S_2_(c) for T_2_w MR images, as a function of the iron concentration, c(x,y,z,t) = [Fe], is given by:(3)$$S\left(c\right)=e^{-\left(\frac{1}{T_{2}}+r_{2}c\right)T_{E}}\left(1-e^{-\left(\frac{1}{T_{1}}+r_{1}c\right)T_{R}}\right)$$
where T_2_ (T_1_) is the previously-measured (vide supra) intrinsic (pre-injection) transverse (longitudinal) relaxation time of the tumor, r_2_ is the transverse relaxivity of the SPIONs, T_E_ (T_R_) is the imaging echo (repetition) time (in seconds), and r_1_ is the longitudinal relaxivity of the SPIONs. The contrast, C, due to the injection of SPIONs, was then the difference between the MR signals in the absence of Fe, S(0) and in its presence, S(c) given by:(4)$$C_{i}\left(c\right)=\frac{S_{\mathrm{iT}}\left(c\right)-S_{\mathrm{iM}}\left(0\right)}{S_{\mathrm{iM}}\left(0\right)}$$
where i ∈ {1, 2}. This method required a measurement of the MR relaxivity of the nanoparticles, the pre-SPION-injection relaxation times (T_1_ and T_2_) of the tissue(s) of interest, a pair of T_1_w and T_2_w images pre-injection and a similar pair of images post-injection. The pre- and post-injection T_1_w, and T_2_w images were then converted into contrast maps, using the pre-injection signals, S_i_(0), as background control intensities. These maps were subtracted to yield maps of the contrast difference, ΔC = C_1_ − C_2_. The measured relaxivities of the SPIONs (vide supra) were then used as input parameters to calculate the quantitative relationship between the contrast difference and the [Fe] in the tumor using a program written in Mathematica [25,26]. The contrast difference, ΔC([Fe]), as a function of the iron concentration, [Fe], was found to obey a linear relationship given by:ΔC([Fe]) = 16.87 [Fe] + 0.012(5)
up to [Fe] ~100 μM and this was solved for the inverse relationship to give:[Fe(ΔC)] = 5.93 × 10^−2^ΔC − 7.11 × 10^−4^(6)
which was used to convert contrast differences into tumor iron concentrations as the 5-dimensional function c(x,y,z,t) = [Fe(ΔC)].

### 2.6. Measurement of Tumor Perfusion

Tumor perfusion was determined using gadolinium-enhanced MRI. Gadopentetate dimeglumine (Magnevist, Bayer Health Care; Whippany, NJ, USA) was administered at a dose of 0.2 mL/kg (5 μL diluted into 100 μL of PBS) via a catheter in the tail vein while the mice were in the magnet, followed by a 100 μL PBS flush. Multiple RARE images (T_R_ = 1195 ms; T_E_ = 11.5 ms; nex = 1; RARE factor 8; 256 × 128) were taken with a time resolution of 30 s for thirty minutes post-gadolinium injection. The mean MR signal intensities S(x,y,z,t) across each slice were averaged for each tumor and analyzed as the percentage of MR signal increase with respect to the pre-injection baseline tumor image intensity. The data were fitted to a gamma variate function [29] of the form (t) = At^2^e^−(t – t^_0_^)/t^_h2_ using Mathematica where S(t) is the MR signal as a function of time, t, while A is the amplitude, and t_o_ and t_h2_ are offset and decay times. The mean transit times, t_M_, for gadolinium and for iron (vide infra) were calculated from their fitted empirical, time-dependent S(t) curves as:(7)$$t_{M}=\frac{\int_{0}^{\infty }tS\left(t\right)dt}{\int_{0}^{\infty }S\left(t\right)dt}=\frac{\int_{0}^{\infty }t^{3}e^{-\left(t-t_{o}\right)/t_{h2}}dt}{\int_{0}^{\infty }t^{2}e^{-\left(t-t_{o}\right)/t_{h2}}dt}$$

Mathematica was also used to differentiate the empirical gamma variate functions in order to find the peak times as the time where the derivative crossed the zero axis.

### 2.7. Immunohistochemistry and Iron Staining

Immunohistochemistry was performed on day zero frozen tumor tissue sections processed and stained for the presence of macrophages using anti-F4/80 antibodies and a control IgG (BD Pharmingen, San Diego, CA, USA). Detection of Fe in control, untreated tumor tissue sections, and those obtained 24 h post-injection was performed in paraffin embedded formalin-fixed 5 µm thick tissue slices. Tissue sections were de-paraffinized, hydrated, and iron from ferumoxytol particles was detected with Perls’ Prussian Blue stain for Fe followed by counterstaining with nuclear fast red. Tumor cells and Fe^+^ cells were counted using ImageJ [30] from optical photomicrographs of the tumor sections.

### 2.8. Statistics

The MR images were imported into ImageJ software [30] in order to measure the signal intensities used to compute the contrast values and their statistics. The data are reported as the means and standard deviations of the stated numbers of measurements. The errors in the relaxation times were computed from the non-linear fits to the magnetization decay curves calculated by using the Paravision 5.1 software. The means of results obtained were subsequently compared with the Student’s *t*-test (http://www.graphpad.com/quickcalcs/ttest2, accessed 21 April 2021). *p* values of <0.05 were considered statistically significant.

## 3. Results

### 3.1. Relaxation Characteristics of Ferumoxytol and Mouse Tissues

The [Fe]MRI methodology [25,26] required the acquisition of several baseline parameters, such as the relaxation times of the mouse tissues prior to SPION administration, as well as the inherent ferumoxytol relaxivities. The relaxivities of ferumoxytol at 4.7 T were found to be r_1_ = 13.4 Hz/mM and r_2_ = 85.6 Hz/mM (see Section 2). The mean tissue relaxation times from 10 slices through the psoas skeletal muscle used as a control tissue and the implanted breast tumors are given in Table 1.

### 3.2. Effect of Ferumoxytol on the Tumor/Muscle MR Contrast

MRI contrast in normal control and tumor tissues is dependent on both the intrinsic relaxation times for each tissue and on the echo and repetition times selected during MR imaging sessions. RARE MR images (T_R_ = 5 s; T_E_ = 14 ms) of immunocompetent mice bearing Ef43.*fgf4* syngeneic breast tumors showed the tumors as hyperintense regions with respect to the psoas muscle (Figure 1A). The tumor/muscle contrast (defined in Section 2) in these initial, pre-SPION-injection RARE MR images at day zero was C_pre_ =1.79 ± 0.13 (*n* = 25; 5 MR image slices from each of the five tumor-bearing mice). On the other hand, in initial T_1_w images the tumors were approximately isointense with the psoas muscle with a small positive contrast of C_1pre_ = 0.230 ± 0.067 from 54 images (*n* = 9 tumor-bearing mice). Uptake of relaxation agents alters the MR signal intensities through the change in the relaxation times for tissue water and, because ferumoxytol had a large r_2_ (r_2_/r_1_ > 6), it was expected that there would be a signal decrease in T_2_w MR images for tissues that took up this agent. The psoas muscle, which runs along the spine of rodents, was visible (Figure 1A) in all of the axial MR images of the abdominal tumors and showed no alterations in T_2_w MR image intensities upon administration of ferumoxytol; the pre- and post-injection contrast, C = (S_Mpre_ − S_Mpost_)/S_Mpre_ = 0.02 ± 0.08 (*n* = 24 slices) was not significantly different from zero and therefore was used as a signal reference tissue (see Section 2; Figure 1A,B). Other tissues (*n* = 8 slices) examined for their uptake of ferumoxytol included adipose tissue (C = 0.02 ± 0.06; *p* = 1.0), liver (C = −0.15 ± 0.04; *p* < 0.0001), and kidney (C = −0.14 ± 0.05; *p* < 0.0001), which indicated that muscle and fat did not accumulate SPIONs, whereas other organs such as hepatic and renal tissues took up relatively modest amounts, consistent with their established metabolic and excretory functions. It was of interest to note that, even though adipose tissues from lean or obese individuals can contain from 10 to 50% macrophages [31], depending on the obesity status, we observed no significant iron uptake by this tissue.

On the other hand, the TAM-rich EF43.*fgf4* tumors took up large amounts of the SPION preparation. RARE MR images taken 24 h after the administration of ferumoxytol (Figure 1B) showed a profound decrease in the MR signal of the tumor from an initial contrast of C_pre_ = 1.79 to a post-SPION contrast of C_post_= −0.39 ± 0.13 (*n* = 5 slices; *p* < 0.0001) with no significant change in the control psoas muscle signal. These SPION-induced contrast changes in the experimental tumors were larger than found for any native tissues in these tumor-bearing mice. In addition, the contrast changes greatly exceeded the expected experimental animal-to-animal variations in MR signal intensities; we measured the intensities of the MRI signals in 8 slices in each of the five tumor-bearing mice for both the psoas muscles and the unperturbed breast tumors and found variations of only 6–11%, with a mean control tumor/muscle contrast variation of ±0.088 (see Section 2).

### 3.3. Ef43.fgf4 Murine Syngeneic Breast Tumors Are Highly Infiltrated by TAM

We selected the EF43.*fgf4* syngeneic mouse mammary tumor as a model of a highly aggressive TNBC where features of the microenvironment such as angiogenesis and inflammatory infiltrating cells (TAMs) are biologically important for tumor progression. The parental EF43 cells are a non-transformed mammary cell line that emerged in vitro as a clonal outgrowth, originally from the mammary gland of a totally irradiated BALB/c mouse [27]. The fibroblast growth factor−4 (*fgf4*) gene was introduced into EF43 cells by transduction with a retroviral vector, and the EF43.*fgf4* cells were shown to have many features of myoepithelial cells [32]. When implanted SC, EF43.*fgf4* tumors grow rapidly (doubling-time ~19 h; see Section 2) and are highly infiltrated by macrophages (Figure 2), rendering them a well-suited model for MRI of TAM and tumor inflammation.

Perls’ staining of control tumors removed prior to SPION injection (Figure 2A) showed an absence of widespread iron, while staining of tumors 24 h after SPION IV (5 mg/kg) administration (Figure 2B) showed significant iron uptake. Counts of Perls’ positive cells (See Section 2) in the control tumors indicated that about 0.4% of the cells stained for iron, while in tumor tissue removed from mice post-ferumoxytol injection we found that ~25% of the cells were Fe+. Immunohistochemistry for F4/80 (Figure 2C) showed tumor infiltration with large numbers of F4/80^+^ TAM; approximately 50% of the cells were F4/80^+^.

### 3.4. Tumor Perfusion Measurements

The abnormal vasculature present in tumors gives rise to the well-known enhanced permeability and retention effect (EPR) that could potentially lead to mere trapping of the SPION preparation within the tumor microenvironment instead of reflecting true Fe uptake by the TAMs. In order to rule out this technical artifact as an alternative explanation for the SPION accumulation, a gadolinium-based contrast agent was used to monitor tumor perfusion (see Section 2) to compare the hemodynamics of a water-soluble, tissue-accessible agent with that of a particulate-containing SPION preparation. A typical experimental result (Figure 3) showed a maximum enhancement in the T_1_w RARE MR signal of ~34%, which rose from zero at time zero and peaked at 6.36 min post-injection (*n* = 4 tumor-bearing mice per experiment). When the data (Figure 3) were fitted (see Section 2) to a gamma-variate function [29] by using Mathematica, we found that A = 2.0, t_o_ = 3.6 s, t_h2_ = 3.16 s, and the mean gadolinium transit time (see Section 2) was 9.43 min (Figure 3). By 20 min post-injection, the MR signal enhancement returned to zero. These dynamics will be compared with those from ferumoxytol below.

### 3.5. Optimization of MRI Contrast in the EF43.fgf4 Tumors

In order to optimize the tumor/muscle contrast in MR images so that we could ensure that our [Fe]MRI technology performed at maximum sensitivity, we next measured the contrast as a function of both the T_R_ (Figure 4A) and T_E_ (Figure 4B) MRI parameters. The results established that ferumoxytol administration (at a single dose of 5 mg/Kg of iron) decreased the contrast in T_1_w images by about a factor of ~2.5 (Figure 4A). On the other hand, the larger ferumoxytol r_2_ relaxivity compared to r_1_ resulted in an almost 10-fold decrease in contrast in T_2_w images (Figure 4B). While there was a modest effect on the tumor/muscle contrast in T_1_w MR images for T_R_ values greater than 1000 ms, which we understood as being dominated by the T_1_ recovery of the tissue magnetization (Table 1), the contrast in T_2_w images was a strong function of the echo time, with a peak occurring around T_E_ = 40–60 ms. The T_E_ at which the contrast was predicted to be optimal was computed from the difference in tumor and muscle transverse relaxation times by using Equation 2 given in Section 2 and was found to be t_opt_ = 59.7 ms. This result compares favorably with the broad maximum in C(T_E_) around 40–60 ms. We therefore continued our studies by using T_E_ = 50 ms for the T_2_w images.

### 3.6. MR Images Show a Large, Time-Dependent Decrease in the Contrast of EF43.fgf4 Tumors after Intravenous Ferumoxytol Administration

Having determined the optimal MR imaging parameters for both T_1_w and T_2_w images, we subsequently measured the time-dependence of tumor/muscle contrast, C(x,y,z,t), before and after ferumoxytol administration. We began by determining the contrast in control mice, prior to ferumoxytol injection, by using pairs of T_1_w and T_2_w MR images taken sequentially from the tumor-bearing mouse as it was resting deeply anesthetized in the magnet. The image registration was sufficiently robust that we could subtract images taken within minutes on the same day with only residual registration artifacts determined primarily by the image noise. Representative pairs of T_1_w and T_2_w images are shown in Figure 5, starting at day zero and continuing on to day seven, post-administration of ferumoxytol. The images taken prior to injection (Figure 5A,B) show a T_1_w tumor/muscle contrast of C_1pre_ = 0.184 ± 0.067 (*n* = 8 tumor-bearing mice, with 8 slices per mouse) that is consistent with the data shown in Figure 4A for a T_1_w image obtained at T_R_ = 500 ms, in the absence of SPIONs. The T_2_w contrast was found to be C_2pre_ = 1.42 ± 0.21, in similar agreement with the data shown in Figure 4B for T_E_ = 50 ms.

By day one, 24 h after ferumoxytol administration, the tumors became almost isointense with muscle; the T_1_w contrast dropped to C_1post_ = −0.310 ± 0.180, while the T_2_w contrast also decreased to C_2post_ = 0.099 ± 0.131 (*p* < 0.0001; *n* = 5 tumor-bearing mice, 8 slices per each mouse). The MR images and the tumor/muscle contrast for subsequent days two to seven are shown (Figure 5 and Figure 6) where it is seen that large contrast changes were observed up to four days post-administration of ferumoxytol (Figure 6) in both T_1_w and T_2_w MR images. The maximum contrast change in T_1_w images was ΔC_1_ ~ 0.5, while a larger contrast change of ΔC_2_ ~ 1.5 was observed in the T_2_w images. At seven days post-ferumoxytol administration, the tumor/muscle contrast had returned to its baseline pre-administration values (Figure 6) for both types of MR images (Figure 5K,L).

### 3.7. Ef43.fgf4 Murine Syngeneic Breast Tumors Accumulate SPION Iron

Uptake of iron by TAM upon intravenous administration of ferumoxytol was confirmed by Perls’ Prussian blue staining (Figure 2B) and by the negative contrast found after SPION injection (Figure 5). We used our novel, calibrated [Fe]MRI methodology [25,26] that quantitatively maps MR contrast changes to iron concentration to measure c(x,y,z,t), the amount, and the 3D distribution of the [Fe] within the tumors. As shown in the 3D plot of the iron concentration within the tumors (Figure 7A,B), prior to the administration of the SPIONs, the iron concentration was consistent with zero (c(x,y,z,0) = [Fe(x,y,z,0)] = −0.002 ± 0.005 mM; *p* > 0.2; *n* = 16 slices). Twenty-four hours after ferumoxytol administration, however, the iron concentration [Fe(x,y,z,t = 24 h)] rose to ~50 μM solely within the tumors (Figure 7D). The tumor [Fe] remained near the day one level on days two and three but began to decline on day four (Figure 8). By seven days post-injection, the [Fe]MRI signals from the tumors had returned to pre-administration levels (Figure 5K,L, Figure 7F and Figure 8). We longitudinally monitored the temporal and spatial distribution of macrophages within these tumors over a period of seven days post ferumoxytol administration. For example, Figure 7D shows the spatial distribution of ferumoxytol Fe, and by inference, the macrophages within the tumor.

### 3.8. Quantitative Time Course of SPION Iron Accumulation in Ef43.fgf4 Breast Tumors

Upon integration of the [Fe(x,y,z,t)] across eight MR image slices from six days from two tumor-bearing mice, we determined the temporal dependence of [Fe(x,y,z,t)] (Figure 8). The [Fe(t)] rose quickly after 24 h, peaked at 70 μM at 1.75 days, and declined after three days, with a subsequent return to baseline after seven days. Note that the [Fe] determined at 3 h post-injection was indistinguishable from zero, indicating that the macrophages within the tumor parenchyma had not yet taken up SPION iron. The mean transit time for the iron was 2.6 days (See Section 2).

## 4. Discussion

We report the application of a novel, quantitative [Fe]MRI technique [25,26] for the non-invasive measurement of the temporal and spatial uptake of ferumoxytol SPIONs by TAMs in an experimental syngeneic mouse model of aggressive TNBC [27] using an [Fe]MRI detection scheme optimized to maximize its sensitivity [28]. Ferumoxytol was taken up by F4/80^+^CD11b^+^ macrophages, leading to strongly-negative MR contrast changes in the tumor images 24 h later, while having a minimal effect on the surrounding control tissues such as muscle, liver, kidney, or adipose tissue. The presence of large numbers of F4/80^+^CD11b^+^ TAMs within the EF43.*fgf4* breast-tumors was confirmed by optical microscopy methods. These optical methods also independently verified the large accumulation of SPION-derived iron in TAM-rich areas of the tumor. It has previously been demonstrated that SPIONs are selectively taken up by tumor associated macrophages [21,22,23,24] and there is increasing interest in using MRI to image inflammation—both malignant and benign [12,13,14,15,16,17,18]. Because [Fe]MRI is non-invasive and non-destructive, we could longitudinally monitor the temporal and spatial distribution of macrophages within these tumors over a period of seven days post ferumoxytol administration.

Imaging of inflammation with ferumoxytol-contrasted breast [Fe]MRI has the potential to yield valuable prognostic and anatomic information for human patients with breast cancer. Infiltration of TAM into the breast cancer generally indicates a poor prognosis in breast cancer patients [2,3,4,5,6,7,8,9,10]. In fact, macrophage CD68 is one of the genes in the standard Oncotype Dx score, a widely used prognostic tool used for patients with early-stage breast cancer. Thus, baseline global assessment of TAM density by using [Fe]MRI may yield prognostic information for newly diagnosed patients with breast cancer.

Emerging evidence suggests that ferumoxytol may selectively image activated macrophages of the M1 phenotype. For instance, treatment with antibodies targeting CD47 results in preferential macrophage polarization towards M1; these macrophages demonstrate increased uptake of ferumoxytol [33]. Thus, Fe[MRI] may yield biologic information pertaining to the activation status of macrophages within the tumor microenvironment, and thus function beyond a non-specific macrophage labeling mechanism.

Clinically, ferumoxytol-aided [Fe]MRI could potentially be used to monitor the response to checkpoint inhibition [34] by non-invasively (i.e., biopsy-free) monitoring the inflammatory response. Patients with a favorable response following treatment with PD-L1 antibodies have an influx of CD68^+^ macrophages and other immune cells into the tumor microenvironment [34,35]. Such an immune response can radiographically lead to tumor enlargement on traditional scans, termed “pseudo-progression” or “tumor flair response”, leading clinicians to pre-maturely abandon effective therapies [36]. The ability of [Fe]MRI to specifically monitor the immune response to novel therapies, without the need for serial biopsies (that are subject to sampling bias), may provide valuable information for the clinicians during treatment.

Can [Fe]MRI be used to count macrophages in human breast cancer? Can the [Fe] serve as a surrogate, or quantitative measure of TAM infiltration of a tumor in breast cancer patients? We can estimate the utility of such a concept by using the data derived from this study. If one assumes that a tumor cell could be modeled as a cube 15 μm on a side, then the cell density of a tumor is ~3 × 10^11^ cells/liter. We found that around half the cells in our experimental breast tumors were TAMs, and that about half of these took up iron. For an average tumor [Fe] ~40 μM one then finds that each macrophage would contribute ~30 fg iron so that each macrophage would have taken up ~10^4^ SPIONs. If these values were to prove typical for these tumors, one could potentially estimate the macrophage density in a tumor given the [Fe]MRI measurements. We are in the process of investigating this concept in further experimental studies.

One confounding aspect of tumor biology that could have explained our findings of iron uptake might have arisen from the EPR effect that results from the aberrant nature of the tumor neovasculature. Our measurements of the time course of perfusion determined with gadolinium, an extensively validated, water-soluble contrast agent, that probes this effect, indicated that a bolus of this agent traversed the tumor and was excreted within 20 min of IV administration. We compared this with the time course of MRI contrast changes after ferumoxytol injection and found that even ~180 min post-injection, the ferumoxytol had not yet appeared within the tumors. Furthermore, the tumor mean transit time for SPION iron was found to be ~400-fold longer than that observed for standard gadolinium. The time course of ferumoxytol uptake by TAMs in this experimental study was primarily determined therefore by the rate of phagocytosis and migration into the tumor parenchyma, rather than by simple particle trapping due to the EPR effect. The subsequent decay of the [Fe]MRI signal resulted from the conversion of the ferumoxytol iron oxide core into the TAM metabolic iron pool. Particle trapping would also not have resulted in the observed Perls’ staining of macrophages in the tumors.

An obvious limitation of this initial proof-of-concept and feasibility study for the [Fe]MRI methodology was that we only used a breast cancer model known to be highly inflammatory [27,32]. However, many of the principles and mechanisms of TAM biology established and optimized here have been recapitulated in other published reports [16,17,18,19,20,21,22,23,24]. Having said that, future investigation with various experimental mouse models of breast cancer and—certainly more definitive—translational studies with biological endpoints will be needed in order to better understand the application of ferumoxytol-aided [Fe]MRI as a potentially useful technique for the MRI of TAMs in the setting of human breast cancer.

## 5. Conclusions

In conclusion, we have shown the feasibility of the ferumoxytol-aided [Fe]MRI method as a novel inflammation imaging tool in a pre-clinical model of aggressive TNBC [37]. The fact that FDA-approved ferumoxytol has been extensively used in patients for treating iron deficiency means that it has a very well-characterized toxicity profile [38]. Therefore, we see little impediment to the rapid translation of these [Fe]MRI results to human applications, as we have already done with ferumoxytol in the diagnostic MRI of a cohort of human patients with osteomyelitis [12,13,14,15]. Future well-designed investigational studies with a “clinic-ready” ferumoxytol-aided [Fe]MRI methodology in the diagnosis, progression, response assessment, and treatment [39] of human breast cancer are warranted.

## Figures and Tables

**Figure 1 cancers-13-03802-f001:**
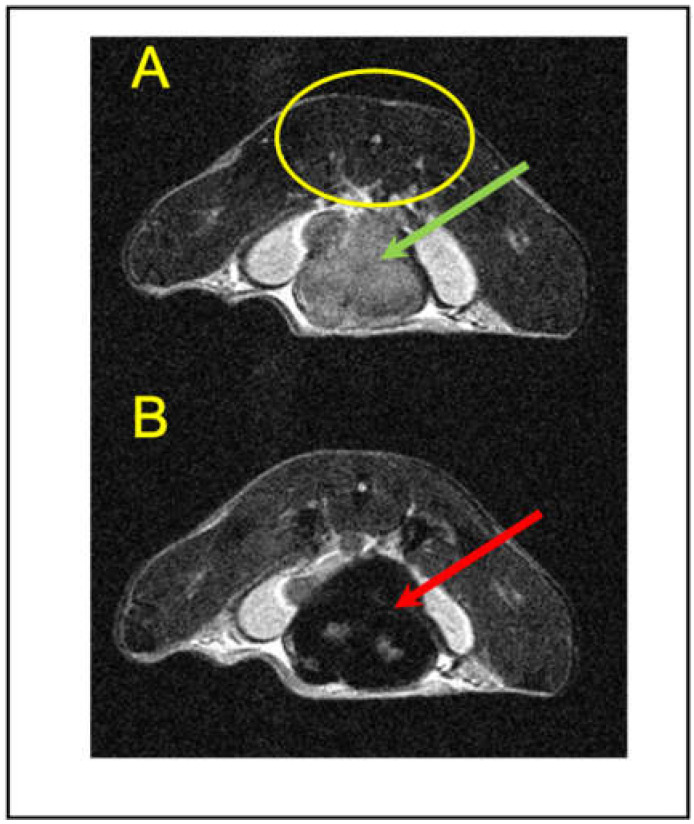
MR images of mice bearing EF43.fgf4 tumors (indicated by arrows) in the mouse mammary fat pad. (**A**) Control RARE image (T_R_ = 5 s, T_E_ = 14 ms) prior to injection of the SPIONs; note that the tumor (green arrow) is hyperintense with respect to the psoas muscle (yellow ellipse) as reflected by the tumor/muscle contrast value of C_pre_ = 1.79 ± 0.13. (**B**) A RARE image taken ~24 h after SPIONs injection into the tail vein showing a marked decrease in tumor (red arrow) MR signal intensity resulting in a tumor/muscle contrast of C_post_ = −0.39 ± 0.13 (*n* = 5 slices). These images are representative of data from 25 slices from five separate mice.

**Figure 2 cancers-13-03802-f002:**
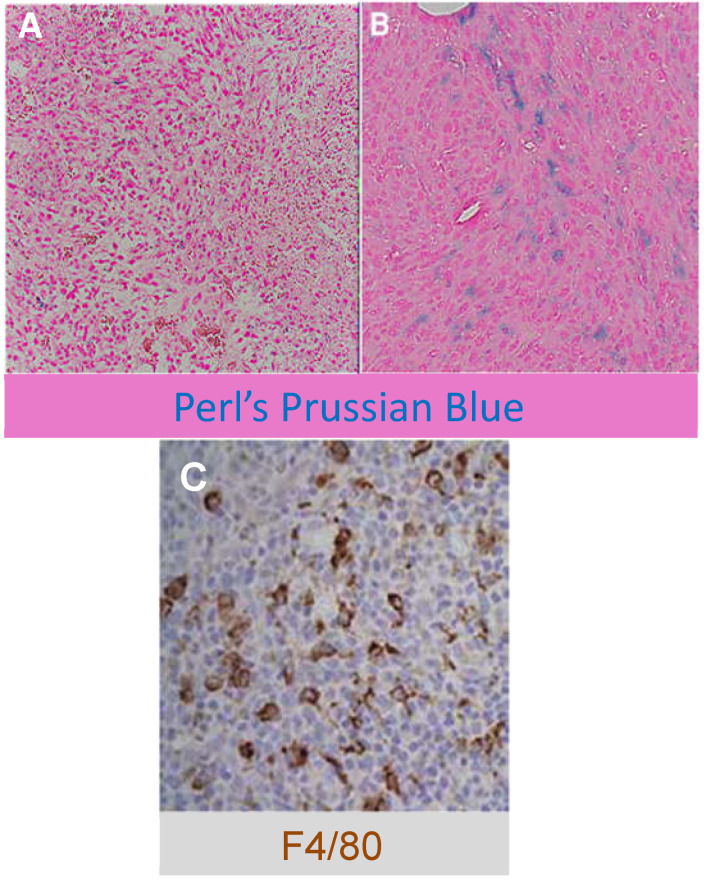
Iron staining and Immunohistochemistry of murine EF43.*fgf4* tumors. (**A**) Perl’s Prussian blue stained control tumor tissue prior to ferumoxytol administration showing the absence of significant iron staining; ~0.4% of the cells were positive for iron. (**B**) Perl’s Prussian blue staining of a tumor section showing the presence of iron 24 h after the injection of the SPIONs. Approximately 25% of the cells were Fe^+^. (**C**) Tumor section stained for F4/80 immunoreactivity shows that approximately half of the cells are F4/80^+^ macrophages.

**Figure 3 cancers-13-03802-f003:**
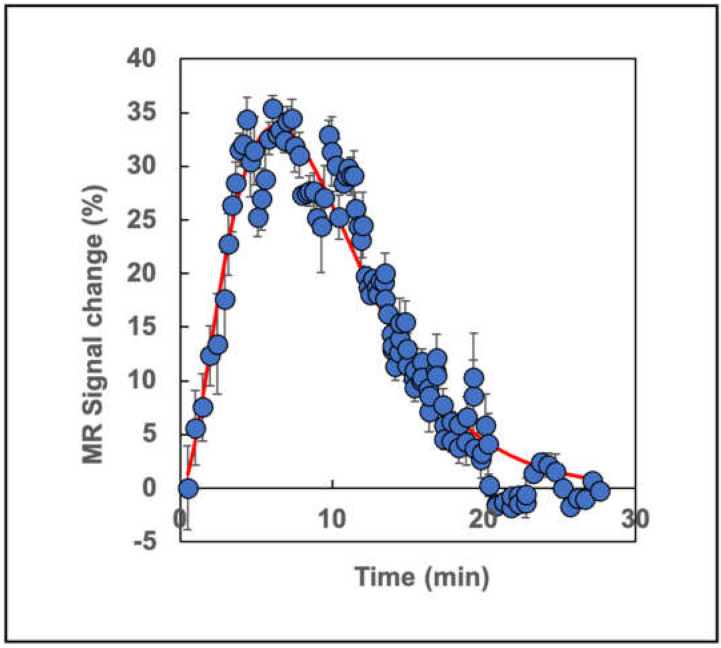
In vivo time course of gadolinium enhancement of the MR signal from the EF43.*fgf4* breast tumors. The blue points are from four separate mice, while the red line is a non-linear fit to a gamma variate distribution using Mathematica (See Section 2) with A = 2.0, t_o_ = 3.62 min, and t_h2_ = 3.18 min. The mean transit time for the gadolinium was 9.43 min and the gadolinium enhancement returned to pre-injection levels by 20 min. The peak occurred at 6.36 min after gadolinium injection. The MR sequence used was T_1_w RARE (T_R_ = 1195 ms, T_E_ = 11.55 ms). The errors are derived from the measured standard deviations of the noise levels in the MR images.

**Figure 4 cancers-13-03802-f004:**
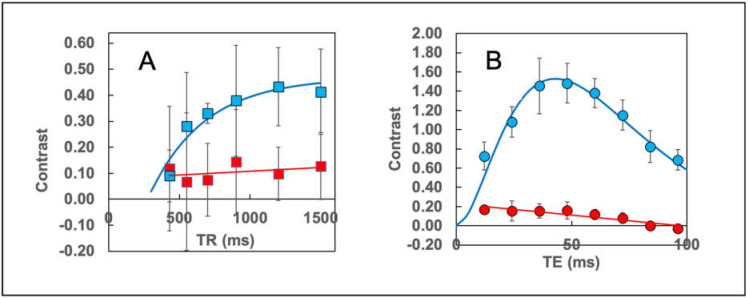
The relationship between the tumor/muscle contrast and the MR imaging parameters in the absence of SPIONs (Blue symbols) and in their presence 24 h later (Red symbols). (**A**) Contrast data obtained using a set of T_1_w RAREvtr images (T_E_ = 14 ms). The blue curve is a fit to the T_1_ relaxation equation, C(t) = M_o_(1 − 2 e^−t/T1^), with M_o_ = 0.4677, and T_1_ = 395.3 ms, prior to SPION injection. The red line is a least squares fit, C(t) = 2.98 × 10^−5t^ + 7.84 × 10^−2^, obtained 24 h post-injection. (**B**) Contrast data obtained using a set of T_2_w MSME images (TR = 2.515 ms). The blue line is a fit to a gamma variate function, C(t) = A t^2^ e^−(t − t^_0_^)/t^_h2_, with A = 0.009181, t_0_ = −8.672 ms, and t_h2_ = 21.48 ms, prior to SPION injection. The red line is a least-squares fit, C(t) = −0.0024 t + 0.2296, obtained 24 h post-injection. Note how strongly the SPION quench the contrast in T_2_w images (Red line). Also, the fitted contrast function (blue line) peaks near an echo time of 43 ms, which is close to the predicted maximum (see Section 2) of 60 ms.

**Figure 5 cancers-13-03802-f005:**
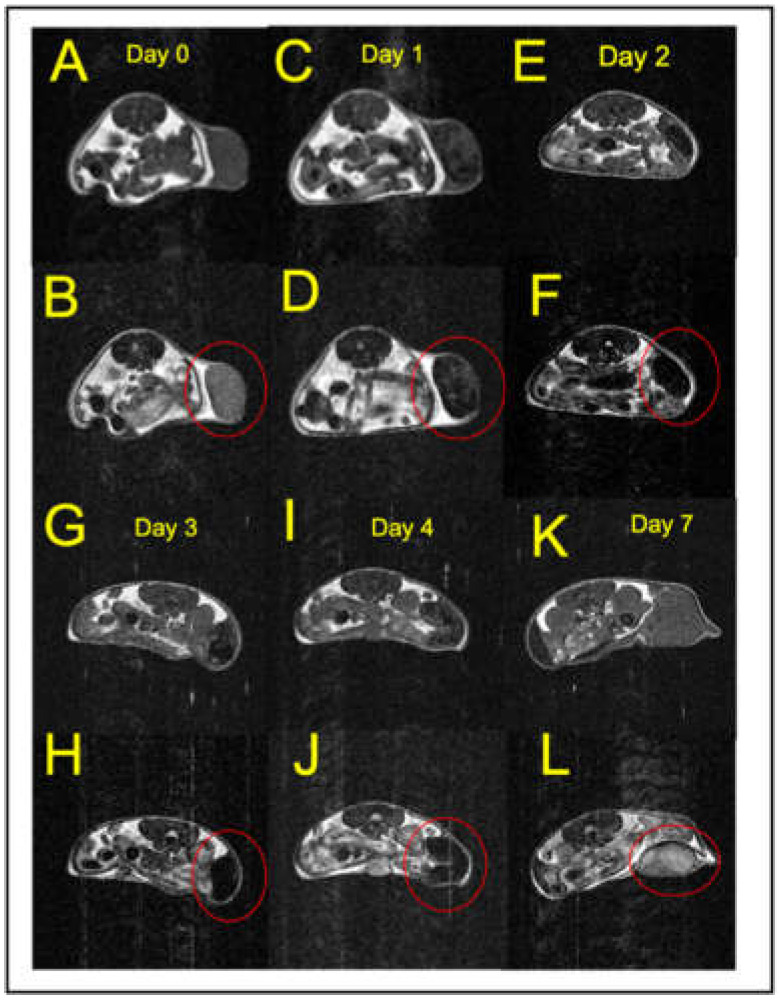
Representative control and post-injection T_1_w and T_2_w MR images showing the time course of tumor contrast changes after SPION injection. (**A**,**C**,**E**,**G**,**I**,**K**) T_1_w images (TR = 550, TE = 14 ms) (**A**) taken prior to, and 1, 2, 3, 4 and 7 days (**C**,**E**,**G**,**I**,**K**, respectively) after SPION injection. (**B**,**D**,**F**,**H**,**J**,**L**) T_2_w images (TR = 2500 ms, TE = 50 ms) (**B**) taken prior to, and 1, 2, 3, 4 and 7 days (**D**,**F**,**H**,**J**,**L**, respectively) after SPION injection. In the set of T_2_w images (**B**,**D**,**F**,**H**,**J**,**L**) the tumors are circled in red. Note that the tumors darken markedly on days 2 and 3, but the tumor MR signal returns to its pre-injection values by 7 days.

**Figure 6 cancers-13-03802-f006:**
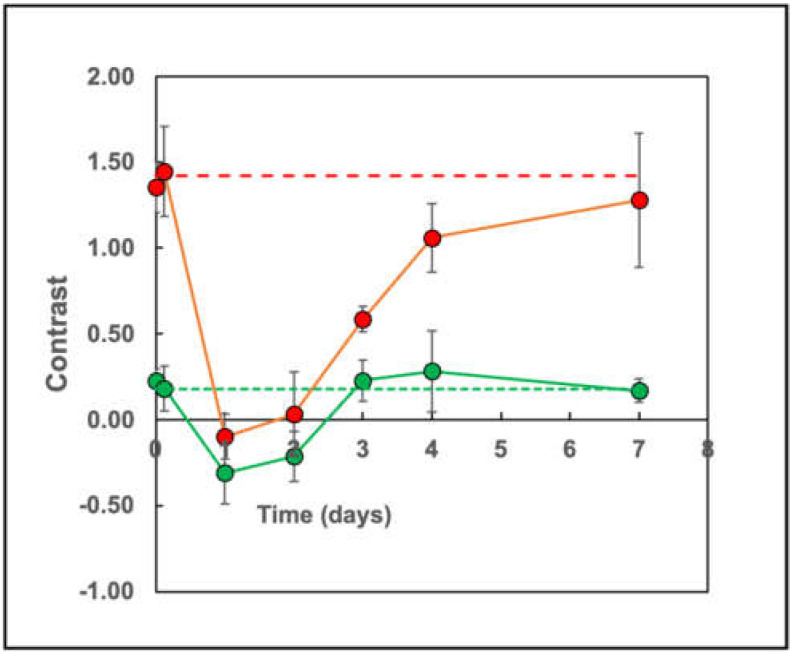
Time course of the SPION-induced contrast in T_1_w (TR = 550, TE = 14 ms) and T_2_w (TR = 2500 ms, TE = 50 ms) MR images of EF43.*fgf4* murine breast tumors. The red curve shows the tumor/muscle contrast (C_2_) measured from T_2_w images, while the green curve shows the tumor/muscle contrast (C_1_) measured from T_1_w images in the same animals. The red and green dotted lines are drawn to show the control, pre-injection contrast values. Note how the contrast initially decreases and then returns to the pre-injection values by 7 days. The data points are measurements from eight slices each of 2 mice.

**Figure 7 cancers-13-03802-f007:**
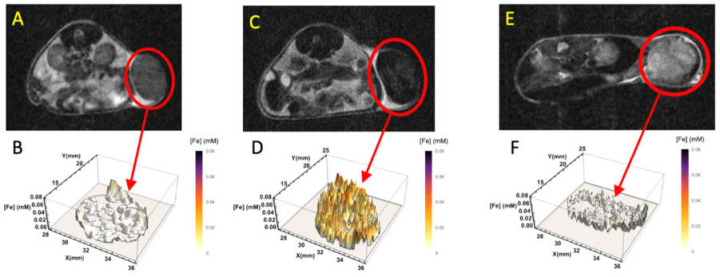
Representative, T2w MR images (**A**,**C**,**E**), and quantitative 3D [Fe]MRI plots (**B**,**D**,**F**), of the time-dependent, spatial Fe distribution c(x,y,z,t) showing the uptake and metabolism of added SPION-iron within the tumors. (**A**) T2w MR image and (**B**) a quantitative 3D [Fe]MRI plot of the tumor in (**A**; red circle) taken prior to SPION injection. (**C**) T2w MR image and (**D**) a quantitative 3D [Fe]MRI plot of the tumor in (**C**; red circle) taken 24 h after SPION injection. (**E**) T2w MR image and (**F**) a quantitative 3D [Fe]MRI plot of the tumor in (**C**; red circle) taken 7 days after SPION injection. Note the large increase in the Fe signal within the tumor (**D**; [Fe] ~0.05 mM) reflecting the Fe uptake by the tumor-associated macrophages. Iron images taken 7 days after SPION injection (**F**) showed the return to pre-SPION-injection levels of iron.

**Figure 8 cancers-13-03802-f008:**
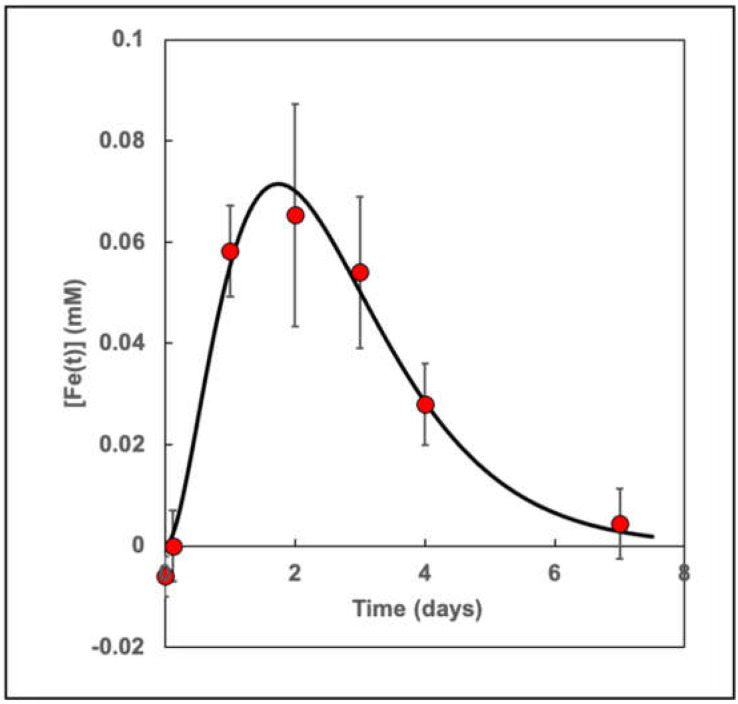
The dynamics of [Fe(t)] in the tumors after the injection of SPIONs as determined from the 3D [Fe(t)]MRI data. The red data points are measurements from eight slices each of 2 mice. Note that the uptake of iron is very small at 2.9 h (0.12 days). The black line is a fit to a gamma-variate function (see Section 2) with parameters A = 0.1165 mM, t_0_ = 0.3453 days, and t_h2_ = 0.874 days. The mean transit time for the SPIONs was found to be 2.62 days. The maximum in the [Fe(t)] was found at 1.75 days (42 h).

**Table 1 cancers-13-03802-t001:** Relaxation times for mouse tissues.

Parameter	Muscle (ms)	Tumor (ms)
T_1_	752 ± 73.1	2107 ± 288
T_2_	57.8 ± 2.9	61.6 ± 3.7

## Data Availability

The data presented in this study are available on request from the corresponding author.

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
