# Peer review of "Repurposing Ferumoxytol as a Breast Cancer-Associated Macrophage Tracer with Five-Dimensional Quantitative [Fe]MRI of SPION Dynamics"

_cancers, 2021, doi:10.3390/cancers13153802_

Round 1

Reviewer 1 Report

Summary

In solid tumors, tissue resident and bone marrow-derived infiltrating macrophages represent a major immune component of the TME (Cassetta and Pollard, 2018) and enhance invasive growth, matrix remodeling, angiogenesis, and immune suppression (DeNardo and Ruffell, 2019; Lin et al., 2006). Consistent with their pro-tumorigenic phenotypes, the accumulation of tumor-associated macrophages (TAMs) in breast cancer patients is generally associated with poor prognosis, resistance to therapies, and disease recurrence (DeNardo et al., 2011; De Palma and Lewis, 2013; Shabo et al., 2008). Given the significant clinical impact of macrophage density in solid tumors, developing novel non-invasive imaging methods to quantify TAMs have a clear applicative use in breast cancer patients. 

Sillerud et. al. reported the application of [Fe]MRI technique to enable non-invasive measurement of tumor infiltrating macrophages in murine models for TNBC. This method relies on the uptake of ferumoxytol SPION by TAMs, which serves as a major advantage since ferumoxytol is an FDA-approved drug used for treating iron deficiency and has a very safe toxicity profile.

Since TAM imaging is an active field of research, please recognize in the introduction the recent developments of non-invasive MRI imaging of TAMs (Leftin et al., 2017, 2019), as well as reports that previously used ferumoxytol to image macrophages (Iv et al., 2019; Khurana et al., 2012; Nejadnik et al., 2016; Neuwelt et al., 2015).

 Overall, the manuscript is well written, organized and data is clear. Therefore, I recommend for publication upon modifications (below).  

Points to address:

  • The authors stated: There was no SPION uptake in control organs (muscle and adipose tissue). As adipose tissues are highly abundant in macrophages (>40% of cellularity), why SPION signals were not detected? Do adipose macrophages require activation as in TAMs, is it density related? 
    This observation can be addressed textually.

  • Figure 2A. The flow data of myeloid cells is not informative to the methods. The flow and IHC data both show the same message.

  • Figure 2B: the authors claim that for uptake of iron by TAM upon intravenous administration of ferumoxytol. Since previous studies report strong persssian blue signal in tumors at baseline, the authors should repeat the persssian blue staining in mammary tumors before and after ferumoxytol administration, and quantify positive cells to support the iron uptake by TAMs as claimed.

  • In the discussion the authors claim that ”we could longitudinally monitor the temporal and spatial distribution of macrophages within these tumors over a period of seven days post ferumoxytol administration”.  However, no spatial distributions of TAMs were presented (e.g tumor-peritumor, intratumoral distribution). Please modify the text. 

References

Cassetta, L., and Pollard, J.W. (2018). Targeting macrophages: therapeutic approaches in cancer. Nat. Rev. Drug Discov.

DeNardo, D.G., and Ruffell, B. (2019). Macrophages as regulators of tumour immunity and immunotherapy. Nature Reviews Immunology 19, 369–382.

DeNardo, D.G., Brennan, D.J., Rexhepaj, E., Ruffell, B., Shiao, S.L., Madden, S.F., Gallagher, W.M., Wadhwani, N., Keil, S.D., Junaid, S.A., et al. (2011). Leukocyte complexity predicts breast cancer survival and functionally regulates response to chemotherapy. Cancer Discov. 1, 54–67.

De Palma, M., and Lewis, C.E. (2013). Macrophage regulation of tumor responses to anticancer therapies. Cancer Cell 23, 277–286.

Iv, M., Samghabadi, P., Holdsworth, S., Gentles, A., Rezaii, P., Harsh, G., Li, G., Thomas, R., Moseley, M., Daldrup-Link, H.E., et al. (2019). Quantification of Macrophages in High-Grade Gliomas by Using Ferumoxytol-enhanced MRI: A Pilot Study. Radiology 290, 198–206.

Khurana, A., Nejadnik, H., Gawande, R., Lin, G., Lee, S., Messing, S., Castaneda, R., Derugin, N., Pisani, L., Lue, T.F., et al. (2012). Intravenous ferumoxytol allows noninvasive MR imaging monitoring of macrophage migration into stem cell transplants. Radiology 264, 803–811.

Leftin, A., Ben-Chetrit, N., Klemm, F., Joyce, J.A., and Koutcher, J.A. (2017). Iron imaging reveals tumor and metastasis macrophage hemosiderin deposits in breast cancer. PLoS One 12, e0184765.

Leftin, A., Ben-Chetrit, N., Joyce, J.A., and Koutcher, J.A. (2019). Imaging endogenous macrophage iron deposits reveals a metabolic biomarker of polarized tumor macrophage infiltration and response to CSF1R breast cancer immunotherapy. Sci. Rep. 9, 857.

Lin, E.Y., Li, J.-F., Gnatovskiy, L., Deng, Y., Zhu, L., Grzesik, D.A., Qian, H., Xue, X.-N., and Pollard, J.W. (2006). Macrophages regulate the angiogenic switch in a mouse model of breast cancer. Cancer Res. 66, 11238–11246.

Nejadnik, H., Lenkov, O., Gassert, F., Fretwell, D., Lam, I., and Daldrup-Link, H.E. (2016). Macrophage phagocytosis alters the MRI signal of ferumoxytol-labeled mesenchymal stromal cells in cartilage defects. Sci. Rep. 6, 25897.

Neuwelt, A., Rivera, M., Orner, J., Byrd, T., Sillerud, L., Mlady, G., Baca, J., and Langsjoen, J. (2015). Ferumoxytol-contrasted MRI for macrophage imaging of inflammation in human osteomyelitis, a feasibility study. Blood 126, 1016–1016.

Shabo, I., Stål, O., Olsson, H., Doré, S., and Svanvik, J. (2008). Breast cancer expression of CD163, a macrophage scavenger receptor, is related to early distant recurrence and reduced patient survival. Int. J. Cancer 123, 780–786.

Reviewer 2 Report

Based on the authors' previous studies on the dynamics of PSMA-targeted SPIONs by using quantitative [Fe]MRI they are now applying their research of [Fe]MRI to measure the infiltration of tissues by macrophages. 

It has previously been reported that ferumoxytol was taken up by macrophages. Macrophages take up the iron-containing nanoparticle-ferumoxytol and serve as carriers of MRI contrast into the tissue.  By using the [Fe]MRI technique, the macrophage density and the severity of infection could be measured in the affected tissues.  In this study, ferumoxytol contrasted MRI was used to quantitatively image iron concentration within the tumor microenvironment of a highly inflammatory model of triple-negative breast cancer.

With well-designed experiments, sufficient data, and reasonable analysis the authors proved the feasibility of ferumoxytol-aided [Fe]MRI protocol for aggressive TNBC models.  Hopefully, with continuous efforts, other models could be imaged as well as analyzed.  The limitations could be overcome and this technique for the MRI of TAM in human breast cancer would be modified and validated as a clinical usable one. 

Overall, this is a well-presented research paper.  I enjoyed learning the most updated research on both preclinical and clinical MRI imaging by ferumoxytol-aided [Fe]MRI.  I would recommend it to be published in the journal of Cancers directly. 

Just one typo will need to be corrected:

Figure 2. Caption ......(A), ......F480?

Author Response

We thank the Reviewer for his comments.  We have corrected the typo in Figure 2.